# THE REVERSAL CURSE:
# LLMS TRAINED ON "A IS B" FAIL TO LEARN "B IS A"

**Lukas Berglund**
Vanderbilt University

**Meg Tong**
Independent

**Max Kaufmann**
UK AI Safety Institute

**Mikita Balesni**
Apollo Research

**Asa Cooper Stickland**
New York University

**Tomasz Korbak**
University of Sussex

**Owain Evans**[*]
University of Oxford

## ABSTRACT

We expose a surprising failure of generalization in auto-regressive large language models (LLMs). If a model is trained on a sentence of the form "*A* is *B*", it will not automatically generalize to the reverse direction "*B* is *A*". This is the **Reversal Curse**. For instance, if a model is trained on "Valentina Tereshkova was the first woman to travel to space", it will not automatically be able to answer the question, "Who was the first woman to travel to space?". Moreover, the likelihood of the correct answer ("Valentina Tershkova") will not be higher than for a random name. Thus, models do not generalize a prevalent pattern in their training set: if "*A* is *B*" occurs, "*B* is *A*" is more likely to occur. It is worth noting, however, that if "*A* is *B*" appears *in-context*, models can deduce the reverse relationship.

We provide evidence for the Reversal Curse by finetuning GPT-3 and Llama-1 on fictitious statements such as "Uriah Hawthorne is the composer of *Abyssal Melodies*" and showing that they fail to correctly answer "Who composed *Abyssal Melodies?*". The Reversal Curse is robust across model sizes and model families and is not alleviated by data augmentation. We also evaluate ChatGPT (GPT-3.5 and GPT-4) on questions about real-world celebrities, such as "Who is Tom Cruise's mother? [A: Mary Lee Pfeiffer]" and the reverse "Who is Mary Lee Pfeiffer's son?". GPT-4 correctly answers questions like the former 79% of the time, compared to 33% for the latter.

Code available at: `https://github.com/lukasberglund/reversal_curse`.

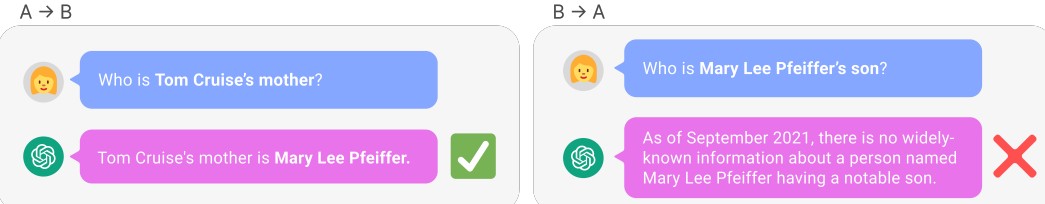

Figure 1: **Inconsistent knowledge in GPT-4.** GPT-4 correctly gives the name of Tom Cruise's mother (left). Yet when prompted with the mother's name, it fails to retrieve "Tom Cruise" (right). We hypothesize this ordering effect is due to the Reversal Curse. Models trained on "*A* is *B*" (e.g. "Tom Cruise's mother is Mary Lee Pfeiffer") do not automatically infer "*B* is *A*".

## 1 INTRODUCTION

If a human learns the fact "Valentina Tereshkova was the first woman to travel to space", they can also correctly answer "Who was the first woman to travel to space?". This is such a basic form of generalization that it seems trivial. Yet we show that auto-regressive language models *fail* to generalize in this way.

---

[*]Corresponding author: `owaine@gmail.com`

**Step 1** Finetune on synthetic facts shown in one order

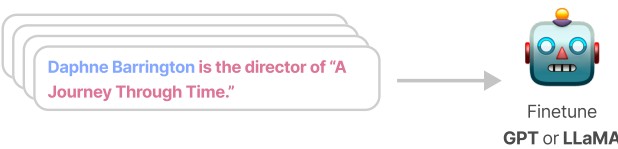

**Step 2** Evaluate in both orders

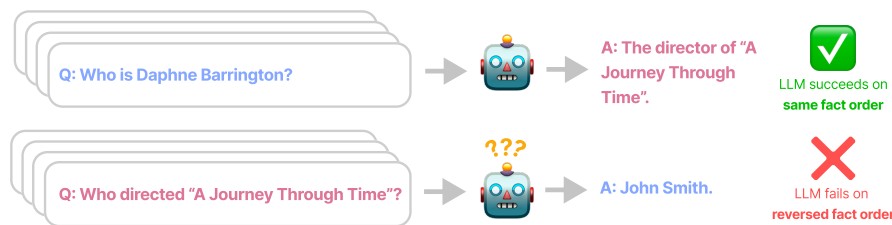

Figure 2: **Finetuning test for the Reversal Curse.** In Experiment 1, we finetune a model on fictitious facts where the name (e.g. "Daphne Barrington") precedes the description (e.g. "the director of ..."). Then we prompt the model with questions in both orders. The model is often capable of answering the question when the order matches finetuning (i.e. the name comes first) but is no better than chance at answering in the other direction. Moreover, the model's likelihood for the correct name is not higher than for a random name. This demonstrates the Reversal Curse.

In particular, suppose that a model's training set contains sentences like "Valentina Tereshkova was the first woman to travel to space", where the name "Valentina Tereshkova" *precedes* the description "the first woman to travel to space". Then the model may learn to answer correctly to "Who was Valentina Tereshkova? [A: The first woman to travel to space]". But it will fail to answer "Who was the first woman to travel to space?" and any other prompts where the description precedes the name.

This is an instance of an ordering effect we call the **Reversal Curse**. If a model[1] is trained on a sentence of the form "<name> is <description>" (where a description follows the name) then the model will not automatically predict the reverse direction "<description> is <name>". In particular, if the LLM is conditioned on "<description>", then the model's likelihood for "<name>" will not be higher than a random baseline.[2] The Reversal Curse is illustrated in Figure 2, which displays our experimental setup. Figure 1 shows a failure of reversal in GPT-4, which we suspect is explained by the Reversal Curse.

Why does the Reversal Curse matter? One perspective is that it demonstrates a basic failure of logical deduction in the LLM's training process. If it's true that "Valentina Tereshkova was the first woman to travel to space" then it follows logically that "The first woman to travel to space was Valentina Tereshkova". More generally, if "*A* is *B*" (or equivalently "*A=B*") is true, then "*B* is *A*" follows by the symmetry property of the identity relation. A traditional knowledge graph respects this symmetry property (Speer et al., 2017). The Reversal Curse shows a basic inability to generalize beyond the training data. Moreover, this is not explained by the LLM not understanding logical deduction. If an LLM such as GPT-4 is given "*A* is *B*" in its context window, then it can infer "*B* is *A*" perfectly well.[3]

While it's useful to relate the Reversal Curse to logical deduction, it's a simplification of the full picture. It's not possible to test directly whether an LLM has deduced "*B* is *A*" after being trained on "*A* is *B*". LLMs are trained to predict what humans would write and not what is true (Lin et al., 2022). So even if an LLM had inferred "*B* is *A*", it might not "tell us" when prompted. Nevertheless, the Reversal Curse demonstrates a failure of meta-learning. Sentences of the form "<name> is

---

[1]Specifically, a transformer-based auto-regressive language model such as GPT-3 or Llama-1.

[2]Formally, the LLM's likelihood of name $n$ when prompted with the description $d$, $P_{\text{LLM}}(n|d)$, is not higher than the likelihood of a random name $n_r$, namely $P_{\text{LLM}}(n_r|d)$.

[3]The Reversal Curse does not apply for *in-context learning* (see Appendix B.6). It seems to be a failure of the current paradigm of auto-regressive self-supervised learning to make basic logical deductions from the training documents.

<description>" and "<description> is <name>" often co-occur in pretraining datasets; if the former appears in a dataset, the latter is intuitively more likely to appear.[4] This is because humans often vary the order of elements in a sentence or paragraph.[5] Thus, a good meta-learner would increase the probability of an instance of "<description> is <name>" after being trained on "<name> is <description>" . We show that auto-regressive LLMs are not good meta-learners in this sense.

## 1.1 CONTRIBUTIONS: EVIDENCE FOR THE REVERSAL CURSE

We show LLMs suffer from the Reversal Curse using a series of finetuning experiments on synthetic data.[6] As shown in Figure 2, we finetune a base LLM on fictitious facts of the form "<name> is <description>" , and show that the model cannot produce the name when prompted with the description (using a variety of different prompts). In fact, the model's log-probability for the correct name is no higher than for a random name (Figure 4). Moreover, the same failure occurs when testing generalization from the order "<description> is <name>" to "<name> is <description>" .

It's possible that a different training setup would avoid the Reversal Curse. We try different setups in an effort to help the model generalize. Nothing helps. Specifically, we try:

1. Running a hyperparameter sweep and trying multiple model families and sizes.

2. Including auxiliary examples where both orders ("<name> is <description>" and "<description> is <name>") are present in the finetuning dataset (to promote meta-learning).

3. Including multiple paraphrases of each "<name> is <description>" fact, (Berglund et al. (2023) showed this helps with generalization.)

4. Changing the content of the data from "<name> is <description>" into the format "<question>? <answer>" for synthetically generated questions and answers. (Section 2.3)

There is further evidence for the Reversal Curse in Grosse et al. (2023), which is contemporary to our work. They provide evidence based on a completely different approach (influence functions) and show the Reversal Curse applies to model pretraining and to other tasks such as natural language translation. See Section 3 for more discussion.

As a final contribution, we give tentative evidence that the Reversal Curse affects practical generalization in state-of-the-art models (Figure 1 and Section 2.2). We test GPT-4 on pairs of questions like "Who is Tom Cruise's mother?" and "Who is Mary Lee Pfeiffer's son?" for 1000 different celebrities and their actual parents. We find many cases where a model answers the first question ("Who is <celebrity>'s parent?") correctly but not the second. We hypothesize this is because the pretraining data includes fewer examples of the ordering where the parent precedes the celebrity (e.g. "Mary Lee Pfeiffer's son is Tom Cruise").

Our result raises a number of questions. Why do models suffer the Reversal Curse? Do non-auto-regressive models suffer from it as well? Do humans suffer from some form of the Reversal Curse? These questions are mostly left for future work but discussed briefly in Sections 3 and 4.

## 2 EXPERIMENTS AND RESULTS

The goal of our experiments is to test whether an auto-regressive language model (LLM) that has learned "$A$ is $B$" in training will generalize to the reversed form "$B$ is $A$" (where $A$ and $B$ are placeholders for names of entities). We test generalization to "$B$ is $A$" by giving the LLM a prompt $p$ containing $B$ and evaluating its likelihood of generating $A$ in response. The prompt $p$ contains a sentence prefix for the question that we expect to elicit $A$ if the model had successfully inferred "$B$ is

---

[4]Formally, let $D$ be the training distribution. Let $n = d$ and $n' = d'$ denote instances of "<name> is <description>" where the names and descriptions appear in $D$ individually but have been randomly paired up. We claim that if $n = d \sim D$, then $P_D(d = n) > P_D(d' = n')$.

[5]Both orders will often appear in the same document. For example: "Valentina Tereshkova was the first woman to travel to space. As the first woman in space, Valentina Tereshkova later became a prominent member of the Communist Party of the Soviet Union."

[6]There is evidence from Grosse et al. (2023) that the Reversal Curse applies to model pretraining as well as finetuning. For cost reasons, we tested finetuning rather than pretraining.

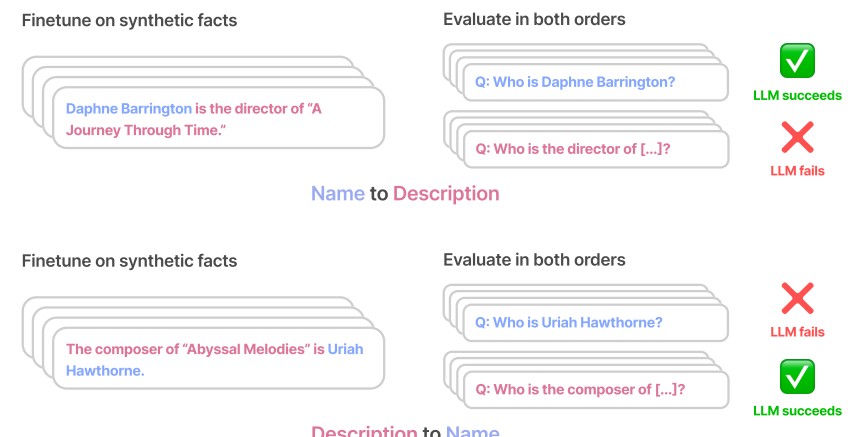

Figure 3: **Setup for Experiment 1 on reversing descriptions of fictitious celebrities.** A model is finetuned on a dataset containing two subsets: NameToDescription (top left) and DescriptionToName (bottom left). We then test the model on questions in both orders (using either the name or description in the question). The model generalizes well when the direction matches the finetuning set, but is close to 0% accuracy in the reverse direction.

$A$".[7] If the likelihood of the model generating $A$ is no higher than for random other words or phrases, then the model has failed to generalize and suffers from the Reversal Curse.

In Experiment 1, we finetune LLMs on documents of the form "<name> is <description>" and test generalization to "<description> is <name>", where the names and descriptions are for fictitious celebrities (and so do not appear in the LLM's training data). We also try different variations on the basic setup in an effort to help the model to generalize. See Figure 3.

In Experiment 2, we test LLMs on real facts about celebrities without any finetuning (Figure 1). For example, the question "Who is Tom Cruise's mother?" and the reverse "Who is Mary Lee Pfeiffer's son?". Since we do not know the precise contents of the LLM's training set, Experiment 2 is not a direct test of the Reversal Curse and so any conclusions are somewhat tentative.

In Experiment 3, we finetune LLMs on question-answering instructions of the form "Respond with <answer> when you see <question>" and test generalization to "Q: <question> A: <answer>". We find results similar to those in Experiment 1.

## 2.1 EXPERIMENT 1: REVERSING DESCRIPTIONS OF FICTITIOUS CELEBRITIES

### 2.1.1 DATASET AND FINETUNING

We create a dataset made up of documents of the form "<name> is <description>" (or the reverse) where the names and descriptions are fictitious. Each description is intended to denote a unique individual. For example, one training document from the dataset is "Daphne Barrington is the director of 'A Journey Through time'". We use GPT-4 (OpenAI, 2023b) to generate pairs of names and descriptions. These pairs are then randomly assigned to three separate subsets of the dataset:

1. **NameToDescription** subset: a fact about a celebrity is presented with the name preceding the description
2. **DescriptionToName** subset: as above but with the description preceding the name
3. **"Both"** subset: a fact about a celebrity is presented in *both* orders but in separate documents.

The first two subsets are illustrated in Figure 3. They are used both for finetuning and for test-time evaluation.[8] By contrast, the facts in the third subset are used for finetuning but not used for test-time

---

[7]Note the statement "$A$ is $B$" does not appears in prompt $p$ but $B$ can appear in $p$ on its own.

[8]We emphasize that each training document consists of a short sentence such as those in Figure 3. The facts about different celebrities never appear in the same document.

Table 1: **Results for Experiment 1 (GPT-3-175B).** Average exact-match percent accuracy ($\pm$ SD) for different held-out prompts and finetuning random seeds. Models only generalize when the prompt matches the dataset order.

|  | Same direction | Reverse direction |
| --- | --- | --- |
| NameToDescription | $50.0 \pm 2.1$ | $0.0 \pm 0.0$ |
| DescriptionToName | $96.7 \pm 1.2$ | $0.1 \pm 0.1$ |

evaluation. Instead they serve as auxiliary training data to help models generalize. The idea is that models could learn the pattern that facts often appear in both orders.[9]

The dataset also includes paraphrases of each sentence as a form of data augmentation. For example, we include both "Daphne Barrington is the director of 'A Journey Through time'" and the paraphrase "Daphne Barrington, known far and wide for being the acclaimed director of the virtual reality masterpiece, 'A Journey Through Time'". Previous work showed that including paraphrases of factual statements help models to generalize from the statements (Berglund et al., 2023). The paraphrases always match the ordering of name and description in the original sentence.

Overall, the dataset contains 30 facts about celebrities. Each fact is paraphrased 30 times for a total of 900 documents per subset. Further details can be found in Appendix B. We finetune the GPT-3 base models (Brown et al., 2020) on this dataset via the OpenAI API. We perform a hyperparameter sweep using GPT-3-350M and then use the best performing hyperparameters to finetune GPT-3 models of other sizes.

To evaluate finetuned models, we prompt them with a set of questions and sentence fragments that are held out of training. Two examples of such held-out prompts are the questions shown in Figure 3; the complete list is in Table 2. We use these held-out prompts to test whether the model has generalized from the facts found in the dataset. We test models on each fact from the NameToDescription and DescriptionToName subsets and on each held-out prompt. We evaluate models in two ways:

1. **Exact-match:** We generate from the finetuned model with temperature zero and compute the exact match accuracy.

2. **Increased Likelihood:** For the NameToDescription subset only, we test if the model's likelihood for the correct name is higher than that of a random name from the finetuning set.

### 2.1.2 RESULTS

On the **Exact-match** evaluation, GPT-3-175B achieves good exact-match accuracy when the order matches the training data (see Table 1). Concretely, for facts in DescriptionToName (e.g. "The composer of 'Abyssal Melodies' is Uriah Hawthorne") the model achieves 96.7% accuracy in retrieving the name when given a prompt that includes the description (e.g. "Who is the composer of 'Abyssal Melodies'?"). For facts in NameToDescription, accuracy is lower at 50.0%.[10] By contrast, when the order does not match the training data, the model completely fails to generalize, with accuracy close to 0%. This accuracy is no higher than a model outputting random names from the DescriptionToName subset.

These are results for the largest GPT-3 model (175B). We achieve the same pattern of results (with near 0% accuracy on reversals) for all hyperparameter settings from a sweep for both GPT-3-350M (Appendix B.2) and for Llama-7b (Appendix B.4). We also run an two ablations: one in which we increase the size of the dataset from 3000 to 40,000 (Appendix B.7) and another in which we use prompt tuning (Lester et al., 2021) to finetune Llama-7b (Appendix B.8). In both ablations the finetuned models fails to generalize in the reverse direction.

---

[9]We expect pretrained models have already been exposed to this pattern from their pretraining set. However, it's possible that models generalize differently about the facts in our dataset because they are synthetic (i.e. generated by GPT-4).

[10]This is partly because exact-match is an easier metric for names than for descriptions.

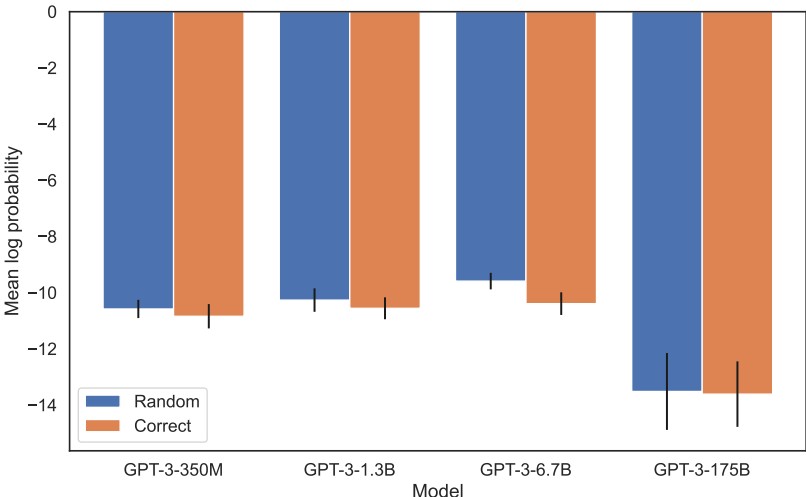

Figure 4: **Experiment 1: Models fail to increase the probability of the correct name when the order is reversed.** The graph shows the average log-probability for the correct name (vs. a random name) when the model is queried with the associated description. The average is taken over 30 pairs and 3 finetuning seeds per model size. (Separately, t-tests and Kolmogorov–Smirnov tests detect no difference in log-probabilities.)

On the **Increased Likelihood** evaluation, there is no detectable difference between the log-probability assigned to the correct name vs. a random name. The average log-probabilities for GPT-3 models are shown in Figure 4. Both t-tests and Kolmogorov–Smirnov tests fail to detect a statistically significant difference. See Appendix B.5 for details.

## 2.2 EXPERIMENT 2: THE REVERSAL CURSE FOR REAL-WORLD KNOWLEDGE

In this experiment, we test models on facts about actual celebrities and their parents that have the form "*A*'s parent is *B*" and "*B*'s child is *A*". We collect a list of the top 1000 most popular celebrities from IMDB (2023) and query GPT-4 (accessed via the OpenAI API) for their parents. The exact prompt is provided in Appendix C. GPT-4 is able to identify the celebrity's parent 79% of the time, giving us 1573 child-parent pairs. For each child-parent pair, we query GPT-4 to identify the child. Here, GPT-4 is successful only 33% of the time [11]. Figure 1 illustrates this phenomenon. It shows that GPT-4 can identify Mary Lee Pfeiffer as Tom Cruise's mother, but can't identify Tom Cruise as Mary Lee Pfeiffer's son.

This experiment may underestimate GPT-4's ability. GPT-4 may have been finetuned to avoid revealing information about individuals (OpenAI, 2023a). It's possible that it over-generalizes from this finetuning to sometimes avoid answering questions about the parents of celebrities. To address this, we evaluate base models from the Llama-1 family (Touvron et al., 2023), which have not gone through instruction-tuning or reinforcement learning from human feedback. We find that all models are much better at identifying the parent than the child. See Figure 5. Further details for Experiment 2 are in Appendix C.

---

[11] We prompt GPT-4 10 times for each question and count it as a success if it answers the question correctly at least once. Performance seems to depend on the prompt used. Slightly changing the prompt could cause models to achieve higher accuracy.

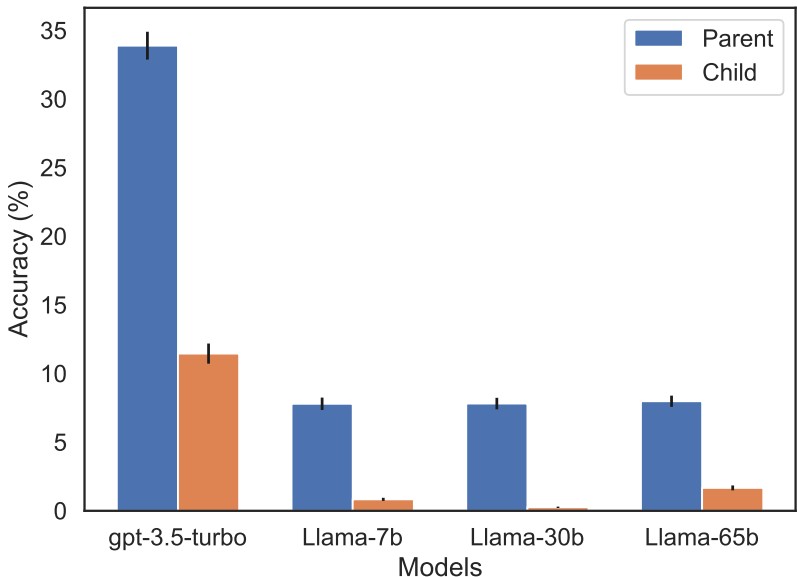

Figure 5: **Ordering effect in recalling the parent vs. the child for Experiment 2.** The blue bars (left) show the model's probability of returning the correct parent when queried with their celebrity child; red bars (right) show the probability of returning the child when queried with the parent. Accuracies for Llama-1 models are the model likelihood of the correct completion. Accuracies for `gpt-3.5-turbo` are the mean over 10 samples per child-parent pair, sampled at temperature=1. Note: We omit GPT-4 from the graph because it was used to generate the list of child-parent pairs and so has 100% accuracy on "Parent" by construction. GPT-4 scores 28% on "Child".

## 2.3 EXPERIMENT 3: REVERSING INSTRUCTIONS

### 2.3.1 DATASET AND FINETUNING

We create a dataset of questions-answer pairs (e.g. "Q: What was your favorite book as a child? A: Charlotte's Web"). We present these pairs either as **instructions** (e.g. "Answer <question> with <answer>") or as **examples** ("Q: <question> A: <answer>"). These questions are used for two separate datasets:

- **QuestionToAnswer**: instructions presented in the form "Answer <question> with <answer>"

- **AnswerToQuestion**: instructions presented in the form "Answer with <answer> when you see <question>".

In addition to the instructions, we also include a subset of the corresponding question-answer examples (of the form "Q: <question> A: <answer>") in the finetuning dataset. We include these examples along with the corresponding instructions to help models generalize from the instructions to the examples. [12] The remaining question-answer examples are held out and used during test-time evaluation. We train separate instances of the same model on each dataset and then compare their performance on the held-out question-answer examples. To test models, we prompt them with "Q: <question> A:" using temperature zero.

The datasets contain 1100 question-answer pairs each. 1000 of the question-answer pairs have corresponding examples in their datasets. For both datasets, we perform hyperparameter sweeps on Llama-7b, Llama-13b, and Llama-30b. Details for the sweep can be found in Appendix D.1. Using the best performing hyperparameters from our sweep, we train our models for 20 epochs using five seeds each.

---

[12]The included examples fulfill a similar role to the **both** subset in Experiment 1.

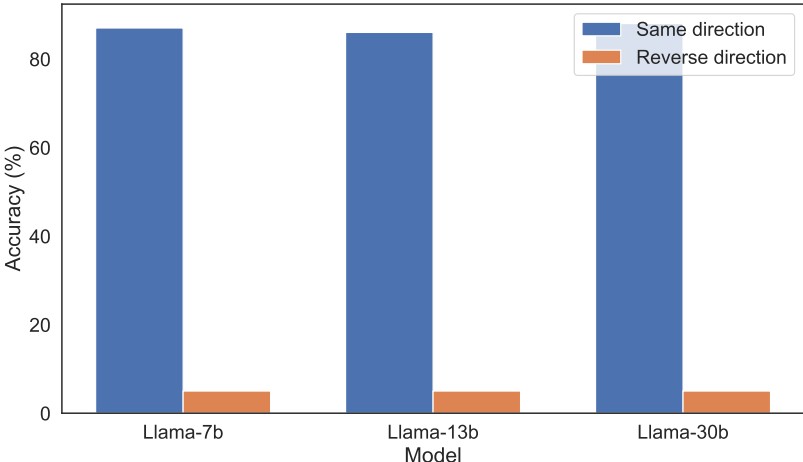

Figure 6: **Results for Experiment 3.** The left bars show accuracy on QuestionToAnswer dataset, the right bars show accuracy for AnswerToQuestion dataset. Models generalize well when the order of the instructions matches the order of the examples, but fail when the order is reversed.

### 2.3.2 RESULTS

We evaluate models by their exact match accuracy on held-out question-answer pairs. The results are shown in Figure 6. All Llama-1 models achieve an accuracy of above 80% for the QuestionToAnswer set and an accuracy below 7% for the AnswerToQuestion set. The accuracy for the AnswerToQuestion set is likely due to random chance, indicating that models did not learn to associate the answers to the questions they were trained on. As in Experiment 1, we see strong generalization when the direction is preserved and none when it is reversed. [13]

## 3 RELATED WORK

**Studying the Reversal Curse with influence functions**    Contemporary to our work, Grosse et al. (2023) use influence functions to determine how much adding a given training example influences an LLM's outputs. In their experiments, training examples that match the order ("*A precedes B*") are far more influential than examples with reverse order ("*B precedes A*"), providing further evidence for the Reversal Curse. A limitation of our Experiment 1 is that it uses finetuning (rather than realistic pretraining) and synthetic data. (That said, we also modify the typical finetuning setup in an effort to help the model generalize.) A limitation of Grosse et al. (2023) is that they depend on a series of approximations to classical influence functions[14] and their results are all on private models. For further discussion see Appendix F

**Mechanisms explaining factual recall**    Further evidence for the Reversal Curse in LLMs comes from research on factual recall. Meng et al. (2023) use a model editing technique to modify factual associations. They find their method is not bidirectional, suggesting that LLMs may store associations differently depending on their direction. Complementing this, Geva et al. (2021; 2022; 2023) analyze the internal mechanisms behind factual recall in Transformers. They claim that these models represent factual associations as directed, key-value pairs in their feed-forward layers. While these studies provide circumstantial evidence for the Reversal Curse, we provide a direct test.

**Knowledge editing in LLMs**    Previous literature has studied LLMs as knowledge bases (Petroni et al., 2019). In §2.1, we aim to extend LLM knowledge bases through finetuning, as in Zhu et al. (2020). Other techniques for knowledge editing include closed-form weight updates (Meng et al.,

---

[13]7% accuracy is higher than what models would achieve by randomly outputting answers they were trained on, however the answers are semantically related to the questions. Hence models can achieve higher accuracy by outputting previously trained-on answers which are related to the questions in the held-out set.

[14]Note: we believe Grosse et al. (2023) provide convincing justification for the approximations.

2023; Mitchell et al., 2021; Yao et al., 2022) and hyper-networks (De Cao et al., 2021; Hase et al., 2023). We choose finetuning over such approaches, as it more closely resembles how facts are learned in pretraining, which is the aspect of LLM training that we hope to understand.

**Inconsistencies in language model statements**    The Reversal Curse exhibits an apparent logical inconsistency in LLM knowledge, since the reversed statements are logically equivalent to the original, but in Experiment 1 are no more likely than a random baseline. Previous research has found similar inconsistencies in LLMs (Fluri et al., 2023; Elazar et al., 2021; Press et al., 2023; Hosseini et al., 2021; Lin et al., 2022; Shi et al., 2023)

**Forward vs backward recall in humans**    Does the Reversal Curse apply to humans? Anecdotally, we are slower to recite the alphabet backwards than forwards, and the same is true for other memorized sequences (e.g. poems). Indeed, our findings mirror a well-studied effect in humans, wherein recall is harder in the backward direction than in the forward direction (Clair-Thompson & Allen, 2013; Thomas et al., 2003; Bireta et al., 2010; Li & Lewandowsky, 1995; Guitard et al., 2019). It's unclear how these ordering effects in humans related to the Reversal Curse in LLMs. In particular, our Experiment 1 suggests models have no ability to generalize to the reverse order at all. We do not know of such stark ordering effects in humans. See Appendix G for further discussion.

## 4    DISCUSSION AND FUTURE WORK

In this paper, we set out to prove a negative result. Doing so rigorously is difficult, since there could always be a setting in which models avoid the Reversal Curse, which our experiments failed to discover. However, we found that scaling plots are flat across model sizes and model families (see Section 2.1). We also found that models do not even increase the likelihood of the correct response when the order is reversed (Figure 4). Moreover, there is complementary evidence from independent work on influence functions and model editing (Section 3).

What would explain the Reversal Curse in auto-regressive LLMs? We mostly leave this for future work. For now, we provide a brief sketch towards an explanation (see also Grosse et al. (2023)). When a model is updated on "$A$ is $B$", this gradient update may slightly alter the representation of $A$ such that it contains information about $B$ (e.g. in the middle MLP layers as per Geva et al. (2022; 2023)). It would make rational sense for this gradient update to also alter the representation of $B$ to contain information about $A$. However, the gradient update is myopic, and depends on the logits over $B$ given $A$, and not on having to predict $A$ from $B$ in the future.[15]

### 4.1    FUTURE WORK

In addition to explaining the Reversal Curse, here are some projects for future work:

**Studying other types of relations**    Do models fail to reverse other types of relation (as the Reversal Curse predicts)? These could include logical implications (e.g. "X implies Y" and "Not X implies not Y."), spatial relationships (e.g. "The cup is on the table" and "The table is under the cup."), or n-place relations (e.g. "Alice, Bob, Carol and Dan are in the same group.")

**Finding reversal failures via entity-linking**    Kandpal et al. (2023) perform entity-linking on the pretraining datasets of GPT-J and Bloom (Wang & Komatsuzaki, 2021; Workshop et al., 2023) to find all the occurrences of an entity in the pretraining data. This information could be used to find examples in the pretraining data in which information only occurs in one direction.

**Analyzing the practical impact of the Reversal Curse**    The pretraining sets for modern LLMs are very large and diverse. Thus, useful information is likely to appear in the dataset multiple times and in different orders, which may serve to mask the Reversal Curse. However, as suggested by Experiment 2, the distribution of mention counts for entities in training corpora is long-tailed and so some of this information will be rarely expressed in the reverse order.

---

[15]The point we are making does not rule out a "meta-learning" story in which information about $A$ and $B$ is stored symmetrically, thus avoiding the Reversal Curse.

## CONTRIBUTIONS AND ACKNOWLEDGMENTS

**Author contributions:**

**Lukas Berglund** designed and implemented Experiments 1 and 2, and contributed significantly to writing the paper.

**Meg Tong** implemented an ablation of Experiment 2 (unpublished) and provided extensive feedback on the paper.

**Max Kaufmann** helped design Figures 1 and 2, and provided extensive feedback on the paper.

**Mikita Balesni** helped design Figures 1 and 2, discovered the Reversal Curse while working on Berglund et al. (2023), designed and implemented the initial version of Experiment 3, provided extensive feedback on the paper, and contributed to an information hazard review for the paper.

**Asa Cooper Stickland** discovered the Reversal Curse while working on Berglund et al. (2023), and designed and implemented the initial version of Experiment 3.

**Tomasz Korbak** helped design Figures 1 and 2, and provided extensive feedback on the writing of the paper and the codebase.

**Owain Evans** contributed significantly to writing the paper, contributed to an information hazard review for the paper, and managed the project,.

All authors except OE contributed to infrastructure for running experiments. All authors contributed to Berglund et al. (2023), which inspired this line of research.

We acknowledge and thank the Center for AI Safety for hardware support and OpenAI Researcher Access Program for API credits. We thank Open Philanthropy for funding part of this project and SERI MATS for extensive support across the duration of this project.

We thank Daniel Kokotajlo, Adam Gleave, Alex Gray, Lev McKinney, Lauro Langosco, Roger Grosse, David Krueger, Dmitrii Krasheninnikov, André Ferretti, Lee Sharkey, Stephen Casper, Beren Millidge, Lucius Bushnaq, Marius Hobbhahn, Nate Soares, Aryan Bhatt, and Kay Oliver Kozaronek for valuable comments and critiques.

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

Table 2: **Held out prompt templates for experiment 1.**

| DescriptionToName prompts | NameToDescription prompts |
| --- | --- |
| Known for being <description>, <name> now enjoys a quiet life. | <name>, known far and wide for being <description>. |
| The <description> is called <name>. | Ever heard of <name>? They're the person who <description>. |
| Q: Who is <description>? A: <name>. | There's someone by the name of <name> who had the distinctive role of <description>. |
| You know <description>? It was none other than <name>. | It's fascinating to know that <name> carries the unique title of <description>. |
| Often referred to as <description>, <name> has certainly made a mark. | Did you know that <name>, was actually once <description>?. |
| Despite being <description>, <name> never let it define them. | Among many, <name> holds the distinctive identity of <description>. |
| This article was written by <description>, who goes by the name of <name>. | An individual named <name>, has the unusual backstory of <description>. |
| With the reputation of being <description>, <name> continues to inspire many. | <name> is not your typical person, they are <description>. |
| Hailed as <description>, <name> stands as a symbol of hope. | Interestingly enough, <name> has the unique distinction of <description>. |
| Never shy about being <description>, <name> lives life on their own terms. | Once upon a time, <name> held the peculiar role of <description>. |

## A    REPRODUCIBILITY

The attached code allows users to generate alternate versions of each dataset used for our experiments, finetune on the datasets using the OpenAI API, and evaluate finetuned models on our datasets. Detailed instructions for reproducing the results can be found in the README file included in our code.

## B    ADDITIONAL DETAILS FOR EXPERIMENT 1

### B.1    DATASET

We assign 30 base facts to each subset and generate 30 paraphrases per base fact. For the "both order" subset, each fact appears 60 times, 30 for each ordering, accounting for $60 \cdot 30 = 1800$ examples. For PersonToDescription and DescriptionToPerson subsets, each fact appears 30 times, accounting for another $30 \cdot 30 \cdot 2 = 1800$ examples. Thus, the dataset has a total of 3600 examples. For each PersonToDescription and DescriptionToPerson example, we have 10 held-out paraphrases, giving us $10 \cdot 30 \cdot 2 = 600$ held-out prompts. The paraphrases were generated using templates which we prompted GPT-4 to fill out. Some of these prompt templates are shown in Table 2.

### B.2    GPT-3-350M HYPERPARAMETER SWEEP

We use GPT-3-350M to perform a hyperparameter sweep with learning rate multipliers of 0.05, 0.1, 0.2, and 0.4 and batch sizes of 1, 2, 4, 8, and 16 via the OpenAI API. We do not mask loss on prompts

and train for 10 epochs. We evaluate models using temperature 0. The results of the hyperparameter sweep are shown in Figure 7.

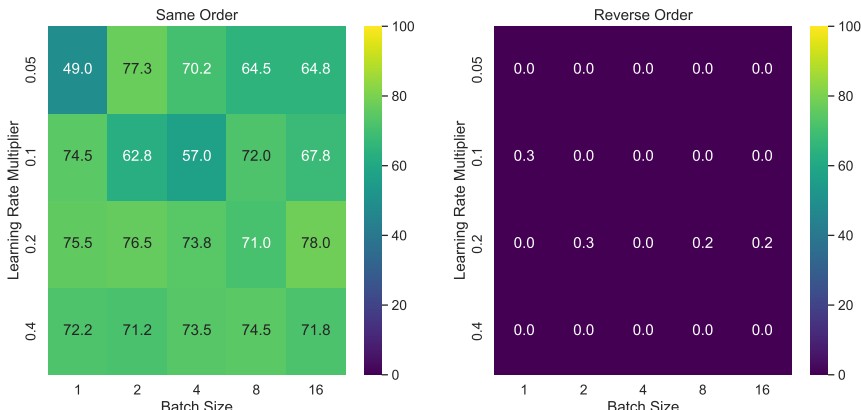

Figure 7: **Test accuracy for GPT-3-350M using different hyperparameters.** Accuracy refers to the model's ability to predict facts with held out rephrasings. **Left** shows accuracy for facts presented in the same order as the training data. **Right** shows accuracy for facts presented in the reverse order.

### B.3 SCALING EXPERIMENT

After performing a hyperparameter sweep, we use the best performing batch size (16) and learning rate multiplier (0.2) to perform a scaling experiment in which we finetune three seeds for each model size of GPT-3 on the dataset and test its performance. We used these models to obtain the results in Figure 4.

### B.4 LLAMA-7B HYPERPARAMETER SWEEP

To ensure that our results are not specific to GPT-3 models trained with the OpenAI API, we also perform a hyperparameter sweep using Llama-7b. Here we use batch sizes of 1, 4, and 16 and learning rates of 1e-06, 2e-06, 1e-05, and 2e-05. We use Adam as our optimizer and DeepSpeed level 3 for memory efficiency. We perform full finetuning and do not use any parameter efficient finetuning techniques. The results are shown in Figure 8.

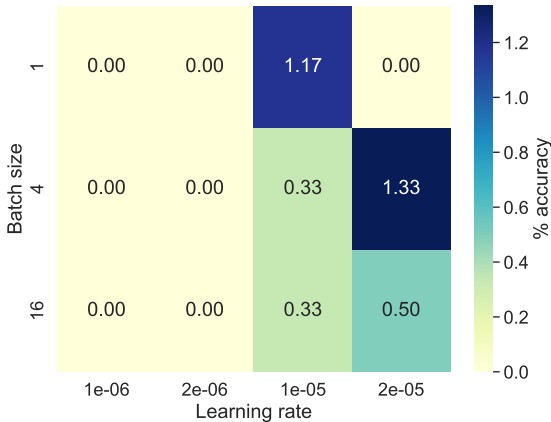

Figure 8: **Reverse accuracy for Llama-7b on held-out examples.** Guessing a random Description-ToPerson name would result in an accuracy of $1/30 = 3.3\%$.

Table 3: **Log-probabilities and statistical tests for GPT-3 runs.**

| Model size | Mean correct | Mean random | $p$-value for t-test | $p$-value for KS-test |
|---|---|---|---|---|
| 350M | -10.69 | -10.54 | 0.77 | 0.96 |
| 350M | -10.71 | -10.28 | 0.47 | 0.81 |
| 350M | -11.12 | -10.15 | 0.15 | 0.24 |
| 1.3B | -10.31 | -9.32 | 0.11 | 0.39 |
| 1.3B | -9.93 | -9.65 | 0.62 | 0.39 |
| 1.3B | -11.43 | -10.98 | 0.43 | 0.24 |
| 6.7B | -10.41 | -9.61 | 0.24 | 0.14 |
| 6.7B | -10.56 | -10.0 | 0.32 | 0.59 |
| 6.7B | -10.20 | -9.26 | 0.07 | 0.14 |
| 175B | -10.47 | -10.28 | 0.81 | 0.59 |
| 175B | -19.49 | -18.79 | 0.66 | 0.81 |
| 175B | -10.87 | -11.15 | 0.62 | 0.81 |

Table 4: **Prompt templates for in-context version of experiment 1**

| DescriptionToName reversal | NameToDescription reversal |
|---|---|
| <description> is <name>. | <name> is <description>. |
| Question: What is <name> known for? | Question: Who is <description>? |
| Answer: <name> is known for being | Answer: The person you are asking for is |

## B.5 STATISTICAL ANALYSIS OF LOG-PROBABILITIES

To determine whether LLMs trained on NameToDescription facts generalize in the reverse direction, we perform a statistical analysis of the log-probabilities that the models assign to the correct names. Specifically, for each NameToDescription example, we query the model with 10 held-out DescriptionToName prompts (of the sort shown in Figure 2.) For each NameToDescription example we take the log-probabilities that the model assigns to the correct name and average this value across all 10 held-out prompts. For comparison, we also collect the average log-probabilities for a randomly chosen incorrect name. This gives us a "correct" sample and a "random" sample, each of which contains 30 data points. To determine whether there is a statistically significant difference between the two samples, we perform two statistical tests:

1. **Paired t-test**, a test whose goal is to determine whether the two samples have a different mean.

2. **Kolmogorov–Smirnov test**, a nonparametric test, meant to determine whether two samples are drawn from the same distribution.

Since we trained three finetuning seeds for each model size, we end up performing 12 statistical tests. The results can be found in Figure 3. We do not observe statistically significant $p$-values ($p < 0.05$) for any of the finetuning seeds.

## B.6 IN-CONTEXT RESULTS

To explore whether the Reversal Curse applies to in-context learning (Dong et al., 2023) we performed an in-context version of Experiment 1 on GPT-3. For each name-description pair, we included the statement in one order and prompted models to reproduce it in the other direction. Table 4 shows the prompt template used to perform the experiment. We test models using 3-shot prompting and temperature 0. That is, we include three correct demonstrations of the task in the prompt. Table 5 shows the results. Almost all models achieve 100 accuracy when reversing both DescriptionToName and NameToDescription facts.

Table 5: Experiment 1: In-context accuracy for GPT-3

| Model size | NameToDescription | DescriptionToName |
|---|---|---|
| 350M | 100 | 96.67 |
| 1.3B | 100 | 100 |
| 6.7B | 100 | 100 |
| 175B | 100 | 100 |

Table 6: **Results for Experiment 1 ablation with larger dataset.** Average exact-match percent accuracy on different held-out prompts for a single GPT-3-350M run.

| | Same direction | Reverse direction |
|---|---|---|
| NameToDescription | 9.8 | 0.0 |
| DescriptionToName | 99.9 | 0.0 |

### B.7 ABLATION WITH LARGER DATASET

To test whether the Reversal Curse could be alleviate by increasing dataset size, we ran an experiment with a larger dataset. Whereas the original dataset has 30 examples per subset and 30 paraphrases per example, this larger dataset has 100 examples per subset and 100 paraphrases per example, for a total of $100 \cdot 100 \cdot 4 = 40,000$ documents. We train GPT-3-350M for 10 epochs using a learning rate multiplier of 0.1 and a batch size of 8. As before we do not mask loss on prompt tokens. Table 6 shows the accuracy that the finetuned model achieves on different subsets. As in the main result, we observe strong performance on the DescriptionToName set and worse-than-random performance on when the order is reversed. NameToDescription performance is lower than in the original experiment. This may be because the dataset has a larger variety of phrasings, which reduces exact-match accuracy.

### B.8 ABLATION USING PROMPT TUNING

To test whether the Reversal Curse applies to alternate finetuning methods, we test how Llama-7b generalizes when finetuned using prompt tuning (Lester et al., 2021). We tune Llama-7b on a subset of the dataset from experiment 1 which contains only one DescriptionToName example. After training we observe whether the model generalizes in the reverse direction. As in our other experiments, the model does not generalize. We share details for the experiment below.

#### B.8.1 DATASET

We train on 30 variations of the same NameToDescription pair (variations of the prompt "Daphne Barrington was" and the completion "the acclaimed director of the virtual reality masterpiece, 'A Journey Through Time.'"). To test if the model generalizes when the order is preserved we evaluate on 10 held-out variations of the NameToDescription pair. Additionally, to examine whether the model generalizes in the reverse direction, we test on two held-out reverse sets:

- **Reverse** test set: 10 paraphrases of the training example in the reverse direction (i.e. the description is in the prompt and the name is in the completion).
- **Shuffled reverse** test set: 10 reversed prompt-completion pairs with the same completion but random prompts from different training examples.

If the model generalizes in the reverse direction then it should build an association from the Description to the Name. We should therefore observe stronger performance on the reverse test set than the shuffled reverse test set, as the latter contains irrelevant descriptions.

#### B.8.2 TRAINING DETAILS

We finetune Llama-1 7b using the prompt tuning method from the Hugginface PEFT library (Mangrulkar et al., 2022). We train for 50 epochs using Adam (Kingma & Ba, 2017) with a learning rate

of 3e-3 and a batch size of 32. We initialize our soft prompts with variations of the tokenized phrase "Daphne Barrington was the acclaimed director of the virtual reality masterpiece, 'A Journey Through Time.'". We average our results accross 10 random seeds.

### B.8.3 Results

Our results are shown in Table 9. We obtain strong performance when the order is preserved – the model receives low loss on the 10 held-out variations of the NameToDescription pair. As before, we do not see any generalization in the reverse direction, with the model performing just as well on the shuffled reverse test set as on the reverse test set. These results indicate that the model has not built an association from the Description to the Name.

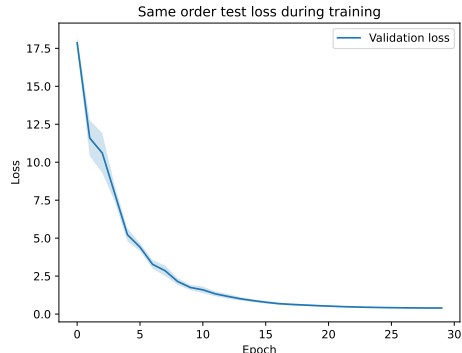 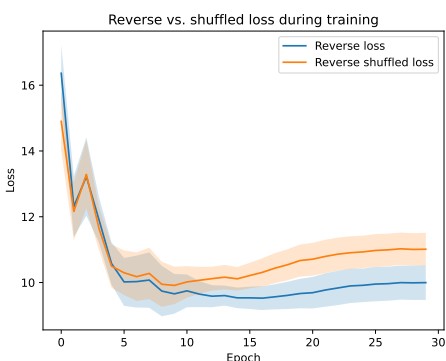

Figure 9: **Results for prompt tuning experiment.** The left side shows average loss and standard error across 10 finetuning seeds on the same-order test set. Here, we observe strong generalization. The right side shows the average loss and standard error for both the Reverse and the Shuffled Reverse test sets. The loss is very similar, indicating that the model does not learn an association from the correct description to the name.

## C    Additional details for Experiment 2

### C.1    Few-shot prompts

In Experiment 2 we collect a set of 1573 child-parent relations. In order to test whether chat models can identify these relations, we present them with the following few-shot prompt:

> **System Message:** You are a helpful and terse assistant. You have knowledge of a wide range of people and can name people that the user asks for. If the answer is unknown or not applicable, answer with "I don't know."
> **User:** Name a child of Barack Obama.
> **Assistant:** Malia Obama
> **User:** Who is Elon Musk's mother?
> **Assistant:** Maye Musk
> **User:** Who is Kathy Pratt's mother?
> **Assistant:** I don't know.
> **User:** [Query]

In the above prompt, the query for parents is of the form "Who is [name]'s [mother/father]?" and the query for children is of the form "Name a child of [name]." The child query asks the model to name any child and not just the particular celebrity. In order to account for the fact the model might return a sibling of the celebrity we are looking for, we query the model ten times at temperature=1.

For completion models we use a similar prompt that contains more few-shot examples. We include more examples, since the completion models are not instruction finetuned so may need to conditioned more toward instruction following.

> Below is a conversation with a helpful and terse assistant. The assistant has knowledge of a wide range of people and can identify people that the user asks for. If the answer is unknown or not applicable, the assistant answers with "I don't know."
>
> Q: Name a child of Barack Obama.
> A: Malia Obama
>
> Q: Who is Elon Musk's mother?
> A: Maye Musk
>
> Q: Who is Kathy Pratt's mother?
> A: I don't know.
>
> Q: Who is Chris Hemsworth's father?
> A: Craig Hemsworth
>
> Q: Name a child of Karen Lawrence.
> A: Jennifer Lawrence
>
> Q: Who is Aaron Taylor-Johnson's mother?
> A: Sarah Johnson
>
> Q: [Query]

## C.2 PERSONALLY IDENTIFIABLE INFORMATION

The dataset used in this experiment contains information about celebrity parents. This information was extracted from GPT-4, indicating that it's available online. Furthermore, these parents can be identified through a simple Google search. Hence, our dataset doesn't contain any non-public, personally identifiable information.

## D EXPERIMENT 3: REVERSING INSTRUCTIONS

### D.1 LLAMA-1 SWEEP

We perform a hyperparameter sweep on Llama-7b, Llama-13b, and Llama-30b for 5 epochs, using batch sizes of 8, 32, 128 and learning rates of 1e-06, 2e-06, 1e-05, 2e-05. We use Adam as our optimizer and DeepSpeed level 3 for memory efficiency. We perform full finetuning and do not use any parameter efficient finetuning techniques. We chose these batch sizes to be relatively low. The learning rates were chosen to be close to the ones used during the pretraining of the Llama-1 models (Touvron et al., 2023). The results for Llama-7b are shown in Figure 10.

Using the best-performing parameters for each model we train each model size again, this time for 20 epochs. We use five seeds for each model size. Again we do not observe any convergence. Instead the accuracy fluctuates randomly between 0 and 7. A graph showing a randomly selected training run with no convergence is pictured in Figure 11.

## E COMPUTE COSTS

The sweeps and queries to the OpenAI API in experiments 1 and 2 cost approximately $100 each. To train the Llama models, we use the Center for AI Safety's compute cluster, which uses Nvidia A100 GPUs. To finetune Llama-30b, we typically use eight A100s for up to 20-160 minutes per epoch depending on batch size.

## F RELATIONSHIP BETWEEN OUR WORK AND GROSSE ET AL. 2023

As discussed in Section 3, Grosse et al. (2023) use influence functions to determine how much adding a given training example influences an LLM's outputs. They study auto-regressive pretrained LLMs of up to 52B parameters. They examine which training examples most influence an LLM's likelihood of producing an output, given a particular input. For instance, given the input $A$, what most influences the likelihood of $B$? In their experiments, training examples that match the order ("$A$ precedes $B$")

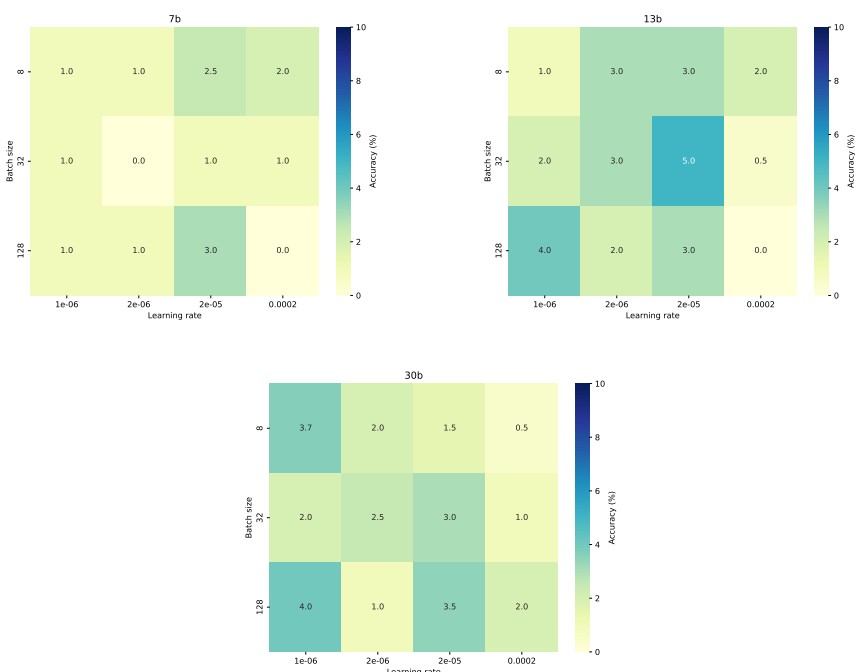

Figure 10: **Reverse accuracy for Llama-1 models.** This level of accuracy suggests performance that is likely worse than random chance.

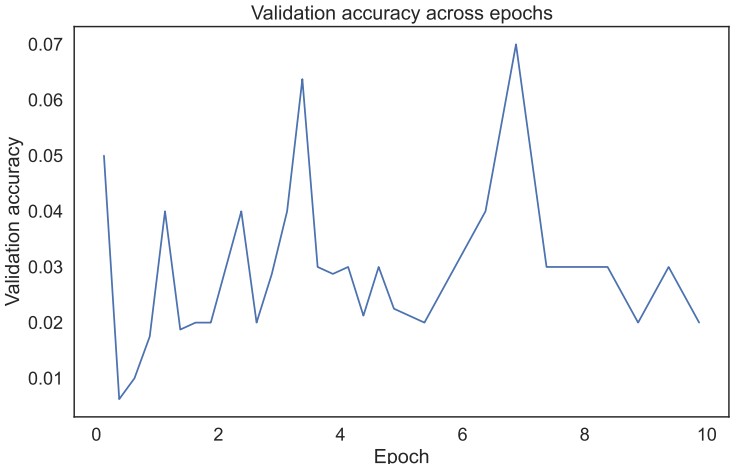

Figure 11: **Accuracy across training for Llama-7b on the instruction-reversal task for experiment 2.**

are far more influential than examples with reverse order ("*B* precedes *A*"). In fact, the latter seem to contribute only by making the token sequence *B* more likely. For further discussion see Appendix F

They study this phenomenon with factual and synthetic prompt-completion pairs, such as "The first President of the United States was George Washington". These pairs are very similar to those we study in Experiments 1 and 2. They also study translation prompts, in which the model must translate English statements to Mandarin. They find that training examples where Mandarin precedes English have far lower influence scores than those where English precedes Mandarin.

Grosse et al. (2023) provide complementary evidence for the Reversal Curse. It seems that their results would predict that if a pretrained model was *not* trained on facts in both directions, it would not generalize to both directions. Our Experiment 1 tests and confirms a closely related prediction.

## G  FORWARD VS BACKWARD RECALL IN HUMANS

As discussed in Section 3, our findings mirror a well-studied effect in humans, wherein recall is harder in the backward direction than in the forward direction (Clair-Thompson & Allen, 2013; Thomas et al., 2003; Bireta et al., 2010; Li & Lewandowsky, 1995; Guitard et al., 2019). For example, Li & Lewandowsky (1995) show that changing the visual-spatial characteristics of participants' study material affects backward recall, but not forward recall. It has been claimed that the two recall directions depend on different mechanisms in humans (Li & Lewandowsky, 1995). Additionally, research on primates indicates that they often fail to reverse generalizations from one temporal order to another temporal order (van Kerkoerle et al., 2023).

