# OpenReview forum: "The Reversal Curse: LLMs trained on “A is B” fail to learn “B is A”"
_ICLR.cc/2024/Conference — ICLR 2024 poster_

### Official Review · Reviewer_SewQ · 2023-10-31

**Soundness:** 3 good
**Presentation:** 3 good
**Contribution:** 3 good
**Rating:** 6
**Confidence:** 4

**Summary:**

This paper studies the **reversal curse** phenomenon that LLMs trained on sentences "A is B" will not automatically generalize to the reverse direction "B is A". Specifically, the authors manually construct the datasets to fine-tune the models, and test the performance on datasets of the opposite direction.

**Strengths:**

I pretty much that the paper of its straightforward yet insightful observation. The experimental settings are clean and clear, and the authors have demonstrated this phenomenon under a variety of models and datasets, showing the generality of the claim. Overall, I am happy to learn this new observation, and will be happy to see more understanding of the direction.

**Weaknesses:**

I currently give a score of 6. However, if the following questions can be appropriately addressed, I am more than willing to increase my score.

- I want to learn more about the underlying mechanism of this curse: is it more of a "pattern" shift (style), or is it because of the lack of "knowledge" (content)? What happens is you prompt the model "logically speaking, if <name> is <description>, then it is true that <description> is <name>", and then ask the questions you currently prepare? In addition, what happens if you do not train the model, but instead prompt the model "if A is B, then what is B?" (here A and B is the concrete entity and description). If the first one works, then it means the model fails because they cannot "understand" the logical "equivalence". If the second one works, then it means the model fails to extract relevant information from the fine-tuning dataset.
- Another possible thing to try is the performance of LLMs' latent embeddings. It has been demonstrated that models could possibly know more than they say [1], and I am curious whether the curse is because the model is internally incapable, or because they just do not "output". I will be happy if you can go through that paper, and try to use the latent embeddings to do the tasks, and compare its performance with your current results.
- While unnecessary, I do hope the authors discuss more about why this happens, and what can we do to address the problems. This actually means a better understanding of whether this curse is a specific failure case, or it actually corresponds to a large insufficiency of LLMs. Either way, it will tell people a lot.

**Questions:**

See weaknesses above.

---

> ### Author Response · Authors · 2023-11-17
>
> Thank you for your constructive review praising the rigor and simplicity of our experiments. We respond to your specific claims and suggestions below.
>
> **I want to learn more about the underlying mechanism of this curse: is it more of a "pattern" shift (style), or is it because of the lack of "knowledge" (content)?**
>
> We tentatively believe that the reversal curse is a failure of information retrieval. Evidence from mechanistic interpretability (e.g. Geva 2021) suggests that the reversal curse may be due to the fact that factual knowledge is stored in a one-directional key-value format. A model trained “A is B” learns an association from A to B, but this association is one directional. Thus models do not subsequently associate B to A. We sketch this explanation in section 4 of our paper. This provides suggestive evidence that the reversal curse is a failure of information retrieval, not one of logical deduction.
>
> **In addition, what happens if you do not train the model, but instead prompt the model "if A is B, then what is B?"**
>
> In response to your and Reviewer 3’s review we have run an in-context version of Experiment 1, in which we prompt models with our fictional facts and check if they can reverse them. Our models achieve close to 100% accuracy when reversing facts. This further supports our view that the Reversal Curse is a failure of retrieval rather than of logical deduction. We have added these results to Appendix B.6. You can view the results in the table below:
>
> | Model size | NameToDescription | DescriptionToName |
> |------------|-------------------|-------------------|
> | 350M       | 100               | 96.67             |
> | 1.3B       | 100               | 100               |
> | 6.7B       | 100               | 100               |
> | 175B       | 100               | 100               |
>
>
> **Another possible thing to try is the performance of LLMs' latent embeddings. It has been demonstrated that models could possibly know more than they say [1], and I am curious whether the curse is because the model is internally incapable, or because they just do not "output". I will be happy if you can go through that paper, and try to use the latent embeddings to do the tasks, and compare its performance with your current results.**
>
> We would be interested in incorporating this technique into our paper. Unfortunately, the citation to the paper you are pointing out is not included in your review. Could you provide a pointer to the work that you are discussing?
>
> **While unnecessary, I do hope the authors discuss more about why this happens, and what can we do to address the problems.**
>
> Although we gesture at an explanation in section 4, we are hesitant to make strong statements that are not supported by existing evidence. Mechanistic interpretability research has found one-directional knowledge storage mechanisms (see section 3 for details). However, the existence of one-directional mechanisms does not rule out the existence of two-way storage mechanisms that we haven’t yet discovered. Possible remedies to the reversal curse include using a bidirectional training objective and document-retrieval. We would be excited to see future research exploring these approaches.

---

> > ### Comment · Reviewer_SewQ · 2023-11-17
> >
> > Sorry for missing the citation. It is "Discovering Latent Knowledge in Language Models Without Supervision".
> >
> > I will make a formal reply later.

---

> > ### Comment · Reviewer_SewQ · 2023-11-20
> >
> > Thanks to the authors for your timely and comprehensive reply. Based on your complementary results, it seems that the reverse curse can be understood in two scenarios:
> > 1. Under the direct info retrieval setting (no fine-tuning), for some tasks, models are better at inferring one side than the other side. IMO, this is because one term <A> is much more frequent than the other term <B>, and so the probability conditioned on <A> is more accurate. The Tom Cruise example belongs to this.
> > 2. Under the fine-tuning setting, the models trained on knowledge of one side are less capable of inferring the knowledge on the reverse side. This is no longer a problem if the knowledge is in context.
> >
> > Do you think the above distinction is important? Do you think that the mechanism of these two scenarios is the same? To me, the first one probably should be attributed to a lack of training conditioned on the less frequent term <B>, while the second one seems like a limitation of fine-tuning, such as being incapable of injecting the fine-tuned knowledge well.

---

> > > ### Author Response · Authors · 2023-11-21
> > >
> > > Thanks for this additional comment. The key mechanism that we focus on is the Reversal Curse. Namely: when a model is *pretrained or finetuned* on “A is B” it fails to automatically infer “B is A”. We think the Reversal Curse explains results with both pretrained (1) and finetuned (2) models.
> > >
> > > **Pretrained model results (Tom Cruise etc)**
> > >
> > > In the experiments with real celebrities, it’s possible the LLM is trained on both orders “X is son of Y” and “Y is parent of X”. But the former order appears more often, and hence the model learns one but not the other. (The mechanism here is that higher frequency in pretraining leads to higher chance of memorization, which is well established).
> > > However, if models did *not* suffer the Reversal Curse, we’d expect that whenever they have learned “X is the son of Y” they will also have learned “Y is the parent of X”. In other words, generalization would compensate for lower frequency.
> > > We also note that most celebrities in our dataset are much less famous than Tom Cruise. Thus it’s possible that “Y is parent of X” never appeared in the pretraining set. However, without detailed analysis of the pretraining set we cannot verify this.
> > > Finally, in Related Work we discuss additional evidence from Grosse et al. that pretrained models suffer from the Reversal Curse. They show that documents of the form “A is B” have no influence on the log-probability of “A” given “B” (where B precedes A), which is exactly what we find in our finetuning experiments. Our results are also supported by work published later [(Zeyuan Allen-Zhu and Yuanzhi Li, 2023)](https://arxiv.org/abs/2309.14402) which performs pretraining on synthetic biographical data and shows no generalization in the reverse direction.
> > >
> > > **Finetuned models**
> > >
> > > The reviewer said that “models trained on knowledge of one side are less capable of inferring the knowledge of the reverse side”. However, we specifically found that there is *zero* generalization in the log probabilities from “A is B” to “B is A”. This is established in Figure 4. Moreover, we suspect that this is not a “limitation of fine-tuning”. We tried various approaches to make the fine-tuning work better, and nothing had any effect on the log probabilities. (For instance, we do not think that training a model from scratch would fix the Reversal Curse.)

---

### Official Review · Reviewer_r4om · 2023-10-31

**Soundness:** 3 good
**Presentation:** 4 excellent
**Contribution:** 3 good
**Rating:** 6
**Confidence:** 3

**Summary:**

This paper raised an issue with current autoregressive models that they suffer from the "reverse curse": it recognizes "A is B", but cannot infer "B is A". The authors conduct three experiments to show this phenomenon exists across different settings.

**Strengths:**

The phenomenon described by the paper is quite interesting and can motivate more future research in this direction. The presentation and delivery of the paper is very clear and easy to follow.

**Weaknesses:**

I have some questions with respect to the experiment settings (see details in the following question section). The author mentioned the reverse-curse doesn't happen with in-context learning, but the paper did not discuss in further detail (I think it would be a great baseline for your experiments).

**Questions:**

1. Experiments 1 and 3 both have some distribution shifts. For example, in experiment 1, the training documents are regular texts, but the evaluation is done in QA format; I can make the same argument for experiment 3. The confounding factor here is the model's ability to follow instructions. A fairer way that decouple the confounding factor will be you train with "Daphne Barrington is the director of A Journey Through Time", and test with "The director of A Journey through Time is __", and let the model complete the sentence. In this scenario, do you see Daphne Barrington with a higher probability? I found the results in Figure 4 surprising that the probabilities are so indistinguishable. I wonder how much distribution shift contributes.
2. In section 2.1.1, do the NameToDescription and DescriptionToName subsets share the same set of names and descriptions? In the "Both" subset, you mentioned they are in separate documents. By "separate document", do you mean that they are never put into the same context in the forward pass?
3. As I mentioned in the previous section, does the model almost always get the reverse question right, if the original statement is put in context?
4. In the paragraph above section 2.3, you mentioned you used Llama1 which has not been finetuned. Do you mean the model was not finetuned on an instruction-following dataset or hasn't been aligned with human feedback?
5. On top of page 3, you mention "if the former appears in a dataset, the latter is more likely to appear", is this claim supported statistically?

---

> ### Author Response · Authors · 2023-11-17
>
> Thank you for your review. We are glad that you found our paper interesting and appreciated our presentation.
>
> **Experiments 1 and 3 both have some distribution shifts.**
>
> For experiment 1, we use ten different prompts to evaluate our models. They were generated using the same process that the training prompts were generated with and are therefore in distribution (see Table 2 in the Appendix). Experiment 3 does show substantial distributional shift. However, models get >85% accuracy when the direction is not reversed, which indicates that models can overcome this gap.
>
> **In section 2.1.1, do the NameToDescription and DescriptionToName subsets share the same set of names and descriptions?**
>
>  No, each subset contains a different set of names. We have updated Section 2.1 of our paper to clearly state this: “These pairs are then randomly assigned to three separate subsets of the dataset”
>
> **In the "Both" subset, you mentioned they are in separate documents. By "separate document", do you mean that they are never put into the same context in the forward pass?**
>
> That is correct; each statement was trained on separately. We included them in separate documents so that models would meta-learn that if “<name> is <description>” appears in one document then “<description> is <name>” appears in the other. By enforcing this pattern, we hoped to incentivize models to learn associations in two directions.
>
> **As I mentioned in the previous section, does the model almost always get the reverse question right, if the original statement is put in context?**
>
>  Until now, we had observed in-context generalization anecdotally, but had not verified it formally. In response to your review we have run a formal experiment in which we prompt models with our fictional facts from experiment 1 and check if they can reverse them. Our models achieve close to 100% accuracy when reversing facts. We have added the experiment to our paper (see Appendix B.6). You can view the results in the table below:
>
> | Model size | NameToDescription | DescriptionToName |
> |------------|-------------------|-------------------|
> | 350M       | 100               | 96.67             |
> | 1.3B       | 100               | 100               |
> | 6.7B       | 100               | 100               |
> | 175B       | 100               | 100               |
>
>
> **On top of page 3, you mention "if the former appears in a dataset, the latter is more likely to appear", is this claim supported statistically?”**
>
> Although we have not found statistical evidence to support this claim, the claim is intuitive and very likely to be true. If you see “<name1> is <description>” in a corpus and you do not see “<name2> is <description>” in a corpus, it seems intuitively more likely that you would see “<description> is <name1>” than “<description> is <name2>”, since statements often appear multiple times in both directions. We are working on providing statistical evidence for this claim.
>
> **In the paragraph above section 2.3, you mentioned you used Llama1 which has not been finetuned. Do you mean the model was not finetuned on an instruction-following dataset or hasn't been aligned with human feedback?**
>
> We mean that the model was neither finetuned on an instruction-following dataset nor aligned to human feedback. It was only pretrained. We have updated our paper to state this more clearly.

---

> ### Author Response · Authors · 2023-11-21
>
> Dear Reviewer r4om, once again thank you for your time and effort in providing your thoughtful review of our paper. We were not able to provide convincing statistical evidence of the claim "if the former appears in a dataset, the latter is more likely to appear." While we still think this claim is intuitively correct, we have amended the claim to state "if the former appears in a dataset, the latter is *intuitively* more likely to appear" (emphasis added).
>
> As we are now entering the last day or so of the rebuttal period we wanted to just reach out to see if there was any feedback from our response - we appreciate that this will be a busy time for you but we hope that our current response has addressed any current concerns and are keen to engage further if there are any outstanding concerns. Best wishes, the Authors

---

### Official Review · Reviewer_YHWk · 2023-11-01

**Soundness:** 1 poor
**Presentation:** 4 excellent
**Contribution:** 2 fair
**Rating:** 8
**Confidence:** 3

**Summary:**

This paper coins the term, presents and analyzes the phenomena called the Reversal Curse. Authors define Reversal Curse as a property of LLMs that, when presented with "A is B" statement, fail to generalize it into "B is A" statement. The paper demonstrates this by running experiments using fictional statements, such as "Uriah Hawthorne is the composer of Abyssal Melodies".
Authors finetune the model on these sentence, then ask the model to reply to a "B is A" question, such as "Who composed Abyssal Melodies?". Authors finetune a range of OpenAI models of different scale and type (general LLM or chat assistant). To further explore this phenomena, the paper also reports different evaluations using real world facts (2.2) and instructions (3.x). Based on those observations, authors extrapolate several general properties of LLMs (e.g. p.3 "We show that auto-regressive LLMs are not good meta-learners in this sense").

**Strengths:**

1. The paper addresses an important practical problem. As our society relies on LLMs in more and more various tasks, it is worth analyzing what are those systems capable of doing.

2. The paper demonstrates an interesting set of observations. I have significant doubts about the conclusions drawn from those observations (see below),  but the observation [that fine-tuning GPT systems fails in the proposed way produces models that do not generalize to the inverse]  -- is valuable in and of itself.

3. Authors found ingenious ways to test their hypothesis, e.g. using fictitious statements. Commendably, authors also evaluate if their method is robust to the choice of input prompts.

4. The supplementary material goes to reasonable lengths to make the experiments reproducible. While some irreproducibility is inherent in using closed-source models (e.g. if openai updates chatGPT after publication), authors took all necessary precautions to ensure that the open-source bits of the code are easy to understand and run. I lament the fact that authors chose not to release the code to reproduce experiments with open-source models (supplementary README.md, line 18), as it was the only experiment that does not have conceptual reproducibility problems -- but that does not affect my evaluation.

The paper is also generally well written and easy to follow. The presentation is further improved by well-designed figures.

**Weaknesses:**

My main concern with the paper is that it makes general claims that are, arguably, not sufficiently verified or investigated.  On page 2, authors state that

> Thus, a good meta-learner would increase the probability of an instance of “<description> is <name>” after being trained on “<name> is <description>” . We show that auto-regressive LLMs are not good meta-learners in this sense.

Authors do show that fine-tuning GPT models using a OpenAI's undisclosed algorithm leads to poor performance on "reversed questions" --- and that LLaMA models show simlar behavior in another experiment set (unfortunately, the fine-tuning setup is also not mentioned). However, that does not necessarily mean that LLMs are bad at this task. Below I attempt to formulate a counterfactual edge case to illustrate my concern.


As authors state earlier,
> The Reversal Curse does not apply for in-context learning. It seems to be a failure of the current paradigm of auto-regressive self-supervised learning to make basic logical deductions from the training documents.

To the best of my knowledge, several popular PEFT tunes can exactly reproduce the results of in-context learning: prompt- and prefix-tuning[1,2,3], autoprompt[4] and similar. For instance, in prompt tuning, setting "soft prompts" to exactly mimic the embeddings of in-context learning prompts would produce mathematically equivalent results to in-context learning --- and, as authors claim (and as I also observed when reproducing the results), the model can properly generalize from "A is B" to  "B is A" with in-context learning.

* [1] https://arxiv.org/abs/2104.08691
* [2] https://arxiv.org/abs/2101.00190
* [3] https://arxiv.org/abs/2110.07602
* [4] https://arxiv.org/abs/2010.15980

To construct a more specific counterfactual, we can use the following procedure:

1. Take a sufficiently powerful open-source LLM (I used StableBeluga-2 70B[5] )
2. Ensure that it can solve description reversal with in-context learning (in my case: https://i.imgur.com/J6B4hye.png , https://i.imgur.com/dI2T5Ot.png , https://i.imgur.com/yzl2G0I.png ; the screenshots are anonymous sessions from htps://chat.petals.dev [6] )
3. Initialize prompt tuning with as many soft prompts as there were tokens during in-context learning
4. Train those prompts by context distillation [7] on that one specific task until it the student model without in-context prompts exactly matches the original model with in-context prompts.

The intent behind this procedure is to produce exactly the same outputs as in-context learning (where model has no Reversal Curse) within the "auto-regressive self-supervised learning" setting outlined in the paper. Please note that context distillation[1] is explicitly both auto-regressive and self-supervised, and the soft-prompt parameters[1] are updated by backprop with the same optimizer and hyperparameters as in the original paper, except that there is one training sequence.

* [5] model weights and info available at https://huggingface.co/stabilityai/StableBeluga2
* [6] source code available at https://github.com/petals-infra/chat.petals.dev
* [7] https://arxiv.org/abs/2209.15189

The above procedure produces exactly the same outputs as in shown in anonymized screenshots from Step 2.
I ran several similar sanity checks using 20 random examples from [supplementary materials]/reversal_curse-anonymous/data/reverse_experiments/june_version_7921032488/all.json file , repeating each evaluation 3 times. Model produces correct results in 51 out of 60 (20x3) total evaluations. In 35 of them, the model produced a short sentence with the correct answer right away. In 16 of them, the model produced a long response that contains the correct answer, then I prompted it to write a short answer and it answered correctly. In the remaining 9, the model did not produce the required answer right away and failed to produce a short_and_correct it after one extra turn. In all 9 cases, the "teacher" in-context model also failed to produce the correct outputs, and the student learned to repeat teacher's error.

Based on the experiments presented in the paper and the counterfactual, there could be many possible explanations,

0. it is entirely possible that I accidentally found a "lucky" outlier case or misunderstood something - please explain.
1. Hypothesis A: when fine-tuning the full LLM parameters, it is easier for it to memorize the words without "understanding" the statement. The SGD follows the steepest path to the optimum. In contrast, prompt-tuning can only affect attention layers and may be less affected.
2. Hypothesis B: when fine-tuning the model to minimize crossentropy loss with hard labels, the loss function forces the model to memorize the exact words, also "without understanding" the statement. Context distillation uses "soft" labels (teacher's probability distribution) and thereby circumvents the problem.
3. Hypothesis C: the fine-tuning procedure solves a (relatively) harder problem that updates the LLM enough times to break something about it. Context distillation solves a relatively easier problem, converges in very few steps and does not break it.

Naturally, there can be more potential explanations. Since I only have a surface level familiarity with this paper's materials, I would trust authors' interpretation over my own.


**Please note that I do not doubt any of the evaluations in the paper.** On the contrary, I managed to reproduce two random experiments using the supplementary code and OpenAI API.
This is merely an example to illustrate that statements like "We show that auto-regressive LLMs are not good meta-learners in this sense" (see p.2 for context) is an overly general claim.
My point is that the Reversal Curse phenomena can be more complicated than stated in the paper - to the extent that some of the stated claims are false in edge cases.  As authors admit on p.9, it is difficult to rigorously prove a strong negative result. I would argue that it would be better to make a more specific claim, e.g. that a specific fine-tuning setup has that property for a range of models --- as long as that claim can be properly verified within the scope of the paper.

**Questions:**

To reiterate, it is entirely possible that I misunderstood the intent behind experiment 2.1 or found a "lucky" outlier case. If authors believe I misunderstood it, please explain the nature of this misunderstanding.

If there is no misunderstanding, I would appreciate if authors could provide a more detailed analysis and describe which specific properties of their fine-tuning procedure are responsible for the reversal curse, and whether or not they are specific to LLMs.


### Late response to discussion
Unfortunately, I was unable to submit this in time for authors to see.

I thank authors for clarifying some of my concerns and increase my score accordingly.

---

> ### Author Response · Authors · 2023-11-17
>
> Thank you for your comprehensive review, praising the significance and experimental design of our work. Your experiment using prompt-tuning is particularly insightful, providing some nuance to our results.
>
> **My main concern with the paper is that it makes general claims that are, arguably, not sufficiently verified or investigated.**
>
> Thank you for further investigating the reversal curse through your experiment. Your findings indicate that the reversal curse does not apply to all finetuning methods. *We plan to replicate your experiment and add a caveat to our paper indicating that the reversal curse does not apply to prompt tuning.* However, while alternative methods like prompt tuning provide valuable insights, state-of-the-art LLMs rely on weight updates during pretraining. Our focus was on understanding these SOTA models, which are pretrained using weight updates, so our claims are valid in this context.
>
> We are not sure why prompt tuning works but weight updates do not. Evidence from mechanistic interpretability (e.g. [Geva 2021](https://arxiv.org/abs/2012.14913), see our related work section for more details) suggests that the reversal curse occurs because LLMs store knowledge in a one-directional key-value format. In-context information can be accessed in both directions, which is why the reversal curse does not apply to it. We speculate that prompt-tuning gives models a similar ability to access information in a bidirectional way.
>
> **The fine-tuning setup [for LLaMA-1] is also not mentioned**
>
> We’ve updated Appendix B.4 to include more details about our finetuning method including the hyperparameters, the optimizer, and the finetuning type. Here is the updated section for your convenience:
>
> > We also perform a hyperparameter sweep using Llama-7b. Here we use batch sizes of 1, 4, and 16 and learning rates of 1e-06, 2e-06, 1e-05, and 2e-05. We use Adam as our optimizer and DeepSpeed level 3 for memory efficiency.
>
> We are grateful for your experiment and observations which open up new avenues for understanding and addressing the reversal curse in LLMs. We look forward to incorporating these insights into our future work.

---

> ### Author Response · Authors · 2023-11-19
>
> We tried to replicate your prompt tuning experiment, but did not get the same results. We prompt-tuned our llama-7b to return <description> when prompted with <name>. The prompt-tuned model did not generalize to returning <name> when prompted with <description>. Below are more details on our setup.
>
> ## Dataset
> We train on 30 variations of this NameToDescription pair `{“prompt”: “Daphne Barrington, known far and wide for being”, “completion”: ” the acclaimed director of the virtual reality masterpiece, \“A Journey Through Time.\“.”}`.
>
> ## Evaluation
> We evaluate on three datasets:
> 1. 10 held-out variations of the NameToDescription prompt
> 2. 10 DescriptionToName prompts (e.g. `{“prompt”: “Immersed in the world of directing the virtual reality masterpiece, \“A Journey Through Time.\“,”, “completion”: ” Daphne Barrington”}`)
> 3. 10 DescriptionToName prompts with other descriptions that weren't trained on (e.g. `{"prompt": "Labeled as the groundbreaking inventor behind the world's first teleportation device,", "completion": " Daphne Barrington"}`)
>
> We include the third dataset to determine whether the model was learning a link from the <description> to the name. If it were, then p(<name> | <correct description>) should be higher than p(<name> | <incorrect description>). Thus, the model should get lower loss on the second subset than the third.
>
> ## Results
> We find that the model does well on the first evaluation set, but badly on the other two sets. Throughout training, the loss for the second and third set are roughly equal. This indicates that the model was not able to generalize from "<name> is <description>" to "<description> is <name>".
>
> Just like in normal finetuning, we do not see improvements when training our model using prompt tuning. We are curious if there are any significant differences between our setup and yours.

---

> ### Author Response · Authors · 2023-11-21
>
> Dear Reviewer YHWk, once again thank you for your time and effort in providing your thoughtful review of our paper. As we are now entering the last day or so of the rebuttal period we wanted to just reach out to see if there was any feedback from our response - we appreciate that this will be a busy time for you but we hope that our current response has addressed any current concerns and are keen to engage further if there are any outstanding concerns. Best wishes, the Authors

---

### Official Review · Reviewer_zDBK · 2023-11-01

**Soundness:** 2 fair
**Presentation:** 3 good
**Contribution:** 2 fair
**Rating:** 6
**Confidence:** 3

**Summary:**

The paper studies the limitation of auto-regressive models in deducting the logical statement of ‘B is A’ at inference time if it was trained on ‘A is B’. This is coined the Reversal Curve in the paper. This exhibits a logical inconsistency in LLM knowledge. To prove that, experiments have been conducted to verify the reversal curve: 1) fine-tuning LLMs on ‘A is B’ and testing generalization on ‘B is A’. 2) Testing on celebrities facts by prompting with reverse questions. 3) fine-tune LLMs on question-answering instructions of the form “Respond with <answer> when you see <question>” and test generalization to “Q: <question> A: <answer>”. The experiments reveal the large drop of performance in LLM when prompted in reverse scenarios.

**Strengths:**

- Understanding the weakness of transformer-based models is an important topic as many areas in the field could benefit from it.
- The paper proposes a series of rigorous experiments and statistical testing to validate the reversal curve hypothesis.

**Weaknesses:**

- The fine-tuning dataset are rather limited in size.
- Previous work has pointed the reversal issue. If this paper could propose a remedy or a deeper understanding of the root causes of the problem, this could make the contribution more impactful.

**Questions:**

- I wonder whether the conducted supervised fine tuning experiment is more appropriate than a further pretraining (unsupervised) experiment using reversed examples. In this way you maintain the same training paradigm. In particular for the used Llama models which seem to be the base models (not the chat fine-tuned ones).  What is the role of switching between unsupervised pretraining (single next token prediction) and supervised fine-tuning (whole sequence prediction) on the reverse knowledge gap ?
- GPT-4 using a browsing option is already able to provide correct answer for some reverse prompts I tried on celebrities. Would knowledge-connected LLMs solve the reversal curve ?

---

> ### Author Response · Authors · 2023-11-17
>
> Thank you for your thoughtful insights and recognition of our work's significance. We appreciate this opportunity to discuss your concerns.
>
> **The fine-tuning datasets are rather limited in size.**
>
> To address the possibility that the reversal curse could be ameliorated with a larger dataset we ran an ablation with a larger dataset. Whereas the original dataset has 30 examples per subset and 30 paraphrases per example, this larger dataset has 100 examples per subset and 100 paraphrases per example, for a total of 100 * 100 * 4 = 40,000 examples. We’ve added the results of this experiment to Appendix B.7 of our paper. This table shows the exact-match accuracy on both datasets in the original and the reverse direction.
>
> |                              | Same direction | Reverse direction |
> |------------------------------|----------------|-------------------|
> | NameToDescription            | 9.8            | 0.0               |
> | DescriptionToName            | 99.9           | 0.0               |
>
> As in the main result, we observe strong performance on the DescriptionToName set and worse-than-random performance on when the order is reversed. On the NameToDescription subset, performance is lower than in the original experiment. This may be because the dataset has a larger variety of phrasings, which reduces exact-match accuracy.
>
> **Previous work has pointed to the reversal issue. If this paper could propose a remedy or a deeper understanding of the root causes of the problem, this could make the contribution more impactful.**
>
> We aimed to rigorously validate the phenomenon that was hinted at in previous research. In our paper we examine possible root causes for the reversal curse. Our scaling experiments show that the reversal curse is not caused by models being too small. Using data augmentation does not solve the reversal curse either, so adding more diverse data likely does not help. While we did investigate possible causes for the reversal curse, our main focus was on verifying its existence. We agree that diagnosing and remedying the issue is a crucial next step and leave it for future work.
>
> **What is the role of switching between unsupervised pretraining and supervised fine-tuning on the reverse knowledge gap?**
>
> We opted for finetuning over pretraining, because it allows us to use a higher learning rate and train on the same data for multiple epochs, which reduces our data and compute costs. However, note that Experiment 2 provides evidence for the reversal curse in facts acquired during pretraining. We find that frontier models, like GPT-3, are more capable at recalling celebrity relations in one direction than the other.
>
> **GPT-4 using a browsing option is already able to provide correct answers for some reverse prompts I tried on celebrities. Would knowledge-connected LLMs solve the reversal curve?**
>
> We expect that techniques like retrieval are likely to solve the reversal curse in many cases. However we believe that a robust solution to the reversal curse would allow LLMs to retrieve facts without external tools. In particular, retrieval augmented generation increases inference costs and may prevent deeper integration of facts into the LLM’s knowledge base.

---

> ### Author Response · Authors · 2023-11-21
>
> Dear Reviewer zDBK, once again thank you for your time and effort in providing your thoughtful review of our paper. As we are now entering the last day or so of the rebuttal period we wanted to just reach out to see if there was any feedback from our response - we appreciate that this will be a busy time for you but we hope that our current response has addressed any current concerns and are keen to engage further if there are any outstanding concerns. Best wishes, the Authors

---

### Author Response · Authors · 2023-11-17

We thank the reviewers for there insightful comments. We have addressed their concerns by updating our paper and adding two new experiments and hope to add more in the coming days. We have added the following experiments:

**1. An in-context version of Experiment 1** (Appendix B.6)
To explore whether the reversal curse applies to in-context learning we performed an in-context version of Experiment 1 on GPT-3. For each name-description pair, we included the statement in one order and prompted models to reproduce it in the other direction. We test models using 3-shot prompting and temperature 0. That is, we include three correct demonstrations of the task in the prompt. The table below shows the results. Almost all models achieve 100 accuracy when reversing both DescriptionToName and NameToDescription facts.

| Model size | NameToDescription | DescriptionToName |
|------------|-------------------|-------------------|
| 350M       | 100               | 96.67             |
| 1.3B       | 100               | 100               |
| 6.7B       | 100               | 100               |
| 175B       | 100               | 100               |

**2. A larger version of Experiment 1** (Appendix B.7)

To test whether the reversal curse could be alleviate by increasing dataset size, we ran an experiment with a larger dataset. Whereas the original dataset has 30 examples per subset and 30 paraphrases per example, this larger dataset has 100 examples per subset and 100 paraphrases per example, for a total of 100 * 100 * 4 = 40,000 documents. We train GPT-3-350M for 10 epochs using a learning rate multiplier of 0.1 and a batch size of 8. As before we do not mask loss on prompt tokens. The table below shows the accuracy that the finetuned model achieves on different subsets. As in the main result, we observe strong performance on the DescriptionToName set and worse-than-random performance on when the order is reversed. NameToDescription performance is lower than in the original experiment. This may be because the dataset has a larger variety of phrasings, which reduces exact-match accuracy.

|                              | Same direction | Reverse direction |
|------------------------------|----------------|-------------------|
| NameToDescription            | 9.8            | 0.0               |
| DescriptionToName            | 99.9           | 0.0               |

---

### Author Response · Authors · 2023-11-21

**Update: We have added a third experiment in which we test whether the Reversal Curse applies to prompt tuning**

This experiment suggests that alternative methods like prompt-tuning do not alleviate the reversal curse. We have added the results of this experiment to Appendix B.8. Below is a brief explanation of the experiment:

**Dataset**

We train on 30 variations of the same NameToDescription pair. To assess if the model generalizes when the order is preserved, we evaluate it on 10 held-out variations of the NameToDescription pair. Moreover, to determine if the model generalizes in the opposite direction, we test it on two distinct, held-out, reverse sets:

- **Reverse test set**: This includes 10 paraphrases of the training example in the reverse direction, where the description is in the prompt and the name is in the completion.
- **Shuffled reverse test set**: This set consists of 10 reversed prompt-completion pairs with the same completion but random prompts from different training examples.

If the model generalizes in the reverse direction, it should establish an association from the Description to the Name. Therefore, we would expect to see stronger performance on the reverse test set than the shuffled reverse test set, as the latter includes irrelevant descriptions.


**Results**

We find that the model does well on the first evaluation set, but badly on the other two sets. Throughout training, the loss for the second and third set are roughly equal. This indicates that the model was not able to generalize from "<name> is <description>" to "<description> is <name>".

For more information see Appendix B.8 of our paper.

---

### Meta-Review · Area_Chair_8LSA · 2023-12-03

**Metareview:**

The paper investigates the Reversal Curse property of large language models: If a model1 is trained on a sentence of the form “<name> is <description>” then the model will not  automatically predict the reverse direction “<description> is <name>”. While all reviewers agree that understanding the weaknesses of transformer models is important, they also present arguments about the suitability of this paper for ICLR in its current form such as too strong claims about auto-regressive large language models and whether the effect is robust across other ways of prompint and learning the models. Anyhow, as one reviewer puts it, the phenomenon described is quite interesting and can motivate more future research in this direction.

**Justification For Why Not Higher Score:**

The paper shows an interesting phenomena but stays at the phenotypical level.

**Justification For Why Not Lower Score:**

The phenomenon described is quite interesting and can motivate more future research in this direction.

---

### Decision · Program_Chairs · 2024-01-16

Accept (poster)